A Miocene pygmy right whale fossil from Australia

Marx Felix G. felixgmarx@gmail.com 1 2 3
Park Travis 2 4
Fitzgerald Erich M.G. 3 4 5
Evans Alistair R. 2 3
1 Directorate of Earth and History of Life, Royal Belgian Institute of Natural Sciences , Brussels , Belgium
2 School of Biological Sciences, Monash University , Clayton , Victoria , Australia
3 Palaeontology, Museums Victoria , Melbourne , Victoria , Australia
4 Department of Life Sciences, Natural History Museum , London , United Kingdom
5 National Museum of Natural History, Smithsonian Institution , Washington , D.C. , United States of America
Young Mark
Electronic publication date: 2018 Jun 22
Publication date: 2018
Volume: 6
Electronic Location ID: e5025
Received 2018 Mar 10; Accepted 2018 May 31
Copyright: ©2018 Marx et al.
Copyright year: 2018
Copyright holder: Marx et al.
License: This is an open access article distributed under the terms of the Creative Commons Attribution License, which permits unrestricted use, distribution, reproduction and adaptation in any medium and for any purpose provided that it is properly attributed. For attribution, the original author(s), title, publication source (PeerJ) and either DOI or URL of the article must be cited.
License URL: https://creativecommons.org/licenses/by/4.0/

Keywords: Neobalaeninae, Cetotheriidae, Evolution, Fossil, Victoria, Caperea, Cochlea, Inner ear

Funding: Marie Skłodowska-Curie Global Postdoctoral fellowship 656010/ MYSTICETI Marie Skłodowska-Curie Individual Fellowship 748167/ECHO This research was supported by a Marie Skłodowska-Curie Global Postdoctoral fellowship (656010/ MYSTICETI) to Felix G. Marx, and a Marie Skłodowska-Curie Individual Fellowship (748167/ECHO) to Travis Park. The funders had no role in study design, data collection and analysis, decision to publish, or preparation of the manuscript.

==============================
Neobalaenines are an enigmatic group of baleen whales represented today by a single living species: the pygmy right whale, Caperea marginata, found only in the Southern Hemisphere. Molecular divergence estimates date the origin of pygmy right whales to 22–26 Ma, yet so far there are only three confirmed fossil occurrences. Here, we describe an isolated periotic from the latest Miocene of Victoria (Australia). The new fossil shows all the hallmarks of Caperea, making it the second-oldest described neobalaenine, and the oldest record of the genus. Overall, the new specimen resembles C. marginata in its external morphology and details of the cochlea, but is more archaic in it having a hypertrophied suprameatal area and a greater number of cochlear turns. The presence of Caperea in Australian waters during the Late Miocene matches the distribution of the living species, and supports a southern origin for pygmy right whales.

Introduction

The living pygmy right whale, Caperea marginata (Gray, 1846), is the smallest living baleen whale, and the only living representative of its lineage (Neobalaeninae). Its ecology and behaviour remains poorly understood, but it stands apart from all other whales in its skeletal structure (Beddard, 1901; Buchholtz, 2011), visual capabilities (Meredith et al., 2013), and even the evolution of its mitochondrial tRNAs (Montelli et al., 2016).

The evolution of pygmy right whales is equally obscure, with their phylogenetic position being controversial (Bisconti, 2015; Boessenecker & Fordyce, 2015; El Adli, Deméré & Boessenecker, 2014; Fordyce & Marx, 2013; Gol’din & Steeman, 2015; Marx & Fordyce, 2016; McGowen, Spaulding & Gatesy, 2009), and their fossil record being all but non-existent. Even though molecular divergence estimates date the origin of neobalaenines to 26–22 Ma (McGowen, Spaulding & Gatesy, 2009; Steeman et al., 2009), fossils are currently limited to just Miocaperea pulchra (Bisconti, 2012) from the Late Miocene of Peru, and two Pleistocene Caperea-like specimens from the Northern Hemisphere (Tsai et al., 2017). Additional Late Miocene material from Argentina (Buono et al., 2014) and Australia (Fitzgerald, 2012) is fragmentary, and uncertainly identified. Cetotheriids are closely related to neobalaenines, and generally more abundant; yet, based on the age of Miocaperea, pygmy right whales diverged from other cetotheriids at least prior to 8 Ma, and probably considerably earlier.

The global dearth of pygmy right whale fossils is striking, and suggests that neobalaenines have either been overlooked or, throughout their evolution, have been rare and/or geographically restricted. Here, we report a new fossil of Caperea from the latest Miocene of southern Australia—the second oldest confirmed neobalaenine, and the oldest record of the genus—and discuss its implications for the evolution and biogeography of the lineage as a whole.

Material and Methods

Collection and photography

The specimen was collected in the first half of the 20th century by G. B. Pritchard, and subsequently formed part of the stratigraphic reference collection at Museums Victoria, Melbourne, until it was identified by FGM and EMGF in 2017. Morphological terminology follows Mead & Fordyce (2009), unless indicated. For the figures, photographs of the specimen were digitally focus stacked in Photoshop CS6.

Scanning technique

The periotic was scanned using the Zeiss Xradia 520 Versa (Zeiss, Oberkochen, Germany) at the Monash University X-ray Microscopy Facility for Imaging Geo-materials (XMFIG) The raw CT data were compiled into a 3D model, and digital endocasts of the cochlea segmented using the visualisation software package Avizo (Version 9.2.0 Standard) (FEI). 3D models of both the periotic and the inner ear are available as Supplemental Information.

Cochlear measurements

Basic measurements of the internal structures of the cochlea were taken using the Measure, Slice and Spline Probe tools in Avizo, following the protocols of Park, Fitzgerald & Evans (2016), and subsequently used to calculate standard ratios (axial pitch and cochlear slope) used to quantify cochlear shape (Ketten & Wartzok, 1990). The radii ratio (and thus also the low frequency hearing limit) could not be calculated, as damage prevented us from obtaining a basal radius. See Park et al. (2017b) for a detailed description of all measurements and procedures.

Results

Systematic palaeontology

Cetacea Brisson, 1762	
Mysticeti Gray, 1864	
Cetotheriidae Brandt, 1872	
Neobalaeninae Gray, 1873	
CapereaGray, 1864	
Caperea sp.	
Figs. 1–4	

Referred material. NMV P233333, partial right periotic preserving the pars cochlearis, body of the periotic, and suprameatal area.

Figure 1 Caperea sp. (NMV P233333), right periotic—photographs.

(A) medial, (B) ventral, (C) dorsal, (D) anterior and (E) posterior view. Photo credit: Felix G. Marx.

Figure 2 Caperea sp. (NMV P233333), right periotic—explanatory line drawings.

Photo credit: Felix G. Marx.

Figure 3 Inner ear of Caperea sp. (NMV P233333).

Digital model reconstructed from microCT data in (A) anterior, (B) lateral, (C) dorsal and (D) vestibular view.

Figure 4 Comparison of Caperea sp. with extant C. marginata.

(A, D) Caperea sp. (NMV P233333) and (B, C) extant C. marginata (NMV C28531) in (A, B) ventral and (C, D) posterior view. Photo credit: Felix G. Marx.

Locality and horizon. Beaumaris Bay, northeast side of Port Phillip Bay, Victoria, southeast Australia, near 37°59′34″S, 145°02′32″E. At Beaumaris, the basal phosphatic nodule bed and overlying ∼7 metres of the shallow marine Sandringham Sandstone (VandenBerg, 2016) produce a rich, albeit largely unpublished, assemblage of marine and rare terrestrial vertebrates (see Fitzgerald & Kool, 2015, and references therein). Although NMV P233333 was originally collected as float, its polished surfaces suggest that it was derived from the phosphatic nodule bed at the base of the Sandringham Sandstone. 87Sr/86Sr ratios from phosphatic intraclasts within the latter horizon yield dates ranging from 6.24–5.38 Ma, with dates of 5.98 and 5.59 Ma within 1 metre above the phosphatic nodule bed (Dickinson & Wallace, 2009). These data suggest that NMV P233333 is latest Miocene (Messinian) in age, between 6.2 and 5.4 Ma.

Description

External anatomy. The new specimen from Australia preserves much of the pars cochlearis, except for its anteromedial corner (Figs. 1 and 2). The surface is abraded, but fractured areas are generally flattened and delimited by a clearly defined rim. The anterior and compound posterior processes are lost, but breakage indicates the shape and size of their respective attachments. Notably, the base of the anterior process is small, suggesting that it was separated from the body of the periotic by a marked constriction.

In medial view (Figs. 1A and 2A), the pars cochlearis is longer (ca 29 mm) than high, with no signs of elongation along its preserved cranial rim. The medial surface is eroded, making it impossible to tell whether a promontorial groove was present. The suprameatal area—specifically, the pyramidal process—is hypertrophied, and forms a broadly triangular protuberance that rises dorsally well above the level of the pars cochlearis. Medially, this protuberance is excavated by a sulcus originating from the dorsal vestibular area.

The mallear fossa is either poorly defined or, probably, worn away. The fenestra cochleae is slightly recessed into the posterior face of the pars cochlearis, but remains clearly visible. The posterior cochlear crest (see Ekdale, Berta & Demere (2011) for a discussion of this term) is small and effectively absent. Abrasion likely somewhat reduced its length, but—judging from the generally good state of preservation in this area—not to a major degree. Posterolateral to the fenestra cochleae, the pars cochlearis somewhat bulges posteriorly, further precluding the presence of a large posterior cochlear crest.

In ventral view (Figs. 1B and 2B), the fenestra vestibuli is large (ca 6.5 mm) and slightly oval. There is no distal opening of the facial canal; instead, the latter is entirely open, and developed as a sulcus running lateral to the fenestra vestibuli and on to the anterior face of the pars cochlearis. Medially and laterally, the distal portion of the canal is sheltered by the pars cochlearis and the base of the anterior process, respectively, suggesting that the lack of a bony floor is genuine and not simply a result of abrasion. The fossa for the stapedial muscle is small, and notably offset from the facial sulcus.

In dorsal view (Figs. 1C and 2C), the dorsal vestibular area is rounded, ca 5.5 mm in diameter, and approximately aligned with the proximal portion of the facial canal and the aperture for the cochlear aqueduct. The crista transversa is robust and widely separates the dorsal vestibular area from the facial canal. The aperture for the cochlear aqueduct is approximately circular (max. diameter 2.5 mm), and located medial to the level of the aperture for the vestibular aqueduct. There is no anteroposterior overlap of the two apertures, and thus no en echelon arrangement as seen in balaenopteroids (Ekdale, Berta & Demere, 2011). The aperture for the vestibular aqueduct is developed as a broad slit (max. diameter 8 mm).

In anterior view (Figs. 1D and 2D), the facial canal extends to the cranial rim of the pars cochlearis. It is unclear whether the canal was entirely exposed in life, or whether its proximal portion was once floored, as seen in C. marginata. Lateral to the facial sulcus, and anterior to the suprameatal area, there is a small area of spongy bone, as seen in C. marginata and a variety of other chaeomysticetes. In posterior view (Figs. 1E and 2E), the fenestra cochleae is large (ca 9 mm), oval, and somewhat oblique relative to the dorsoventral axis of the pars cochlearis. The fenestra cochleae and aperture of the cochlear aqueduct are well separated.

Inner ear. The basalmost quarter turn of the cochlea was partially damaged, but enough remained to enable its reconstruction (Fig. 3). The cochlea completes 2.5 turns and bears a distinct tympanal recess, with the scala tympani being inflated radially along approximately the first half turn of the cochlear canal. In vestibular view, the apical turn is tightly coiled and encloses a small open space. The entire apical turn overlaps the section of the cochlear canal immediately below. The reconstructed portion of the cochlea has a height of 12.89 mm, a width of 16.33 mm, a volume of 856.63 mm3, and a canal length of 52.37 mm; inside the basal turn, the secondary spiral lamina extends for 15.71 mm, equal to approximately 30% of the total length of the cochlear canal. The basal ratio is 0.79, the axial pitch 5.16, and the cochlear slope 0.10.

Discussion and Conclusions

Comparisons

NMV P233333 closely resembles extant C. marginata, but not Miocaperea (Bisconti, 2012), in having a (partially) open facial canal, a nearly absent posterior cochlear crest, and a distinct neck separating the anterior process from the body of the periotic (Fig. 4). In addition, both share the presence of relatively large fenestrae cochleae and vestibuli; the oblique orientation of the fenestra cochleae; a well-developed pyramidal process; and the retention of a small, circular aperture for the cochlear aqueduct that is well separated from the aperture for vestibular aqueduct (Fig. 5).

Figure 5 Comparison of Caperea sp. with extant C. marginata.

(A) Caperea marginata (NMV C28531) and (B) Caperea sp. (NMV P233333), both in dorsal view. Photo credit: Felix G. Marx.

The open facial sulcus and constricted base of the anterior process are unique to Caperea, and strongly suggest referral of NMV P23333 to this genus. The new specimen differs from C. marginata, however, in having a markedly wider crista transversa, a larger pyramidal process, a larger stylomastoid fossa, and a more robust neck of anterior process. The combination of these features may suggest the existence of a separate, more archaic species of Caperea, but the fragmentary nature of the specimen prevents us from drawing firmer conclusions. Notably, a similarly robust pyramidal process also occurs in several cetotheriids, including Kurdalagonus, Mithridatocetus, and Herpetocetus.

A tympanal recess of the cochlea is typical of Caperea and most other plicogulans, but not balaenids and more archaic mysticetes, such as eomysticetids (Park et al., 2017b). Two and a half turns give the cochlea a more archaic aspect than that of C. marginata, and are broadly in line with the number of turns in most other mysticetes (Ekdale, 2016; Ritsche et al., 2018). The height and volume of the cochlea are larger than in most mysticetes, but similar to that of C. marginata. The axial pitch is also most similar to Caperea, whereas the cochlear slope value is higher than in all other mysticetes (Ekdale, 2016; Park et al., 2017a; Park et al., 2017b), and more similar to that of Zygorhiza or even some odontocetes (Park, Fitzgerald & Evans, 2016).

Origins of Caperea

Extant Caperea is restricted to the Southern Hemisphere, which, until recently, had also been the sole source of neobalaenine fossils (Bisconti, 2012; Buono et al., 2014; Fitzgerald, 2012). The recent description of two Pleistocene Caperea-like fossils from Italy and Japan provided the first evidence that pygmy right whales once crossed the equator, likely during brief glacial intervals associated with Northern Hemisphere glaciation (Tsai et al., 2017).

The idea that neobalaenines originated in the Southern Hemisphere and only briefly ventured north is plausible, but so far has remained difficult to test in light of the sparse fossil record. Thus, until now, the two northern occurrences of Caperea constituted the majority of the confidently identified neobalaenine record, contrasted only by Miocaperea pulchra from the Late Miocene of Peru. Our new specimen firms up the ancient southern history of the Caperea lineage, and suggests that neobalaenines may indeed have originated in austral seas (Fig. 6).

Additionally, our identification of an unambiguous neobalaenine periotic from the Sandringham Sandstone at Beaumaris corroborates Fitzgerald’s (2012) account of an isolated, putatively neobalaenine compound posterior process from the same unit and locality. Unlike our new material, the specimen described by Fitzgerald (2012) is notably larger than adult C. marginata, raising the possibility that it represents a distinct taxon. On the other hand, the posterior process of mysticetes dramatically increases in both absolute and relative size during ontogeny (Bisconti, 2001), making it plausible that the two Beaumaris specimens may be congeneric or even conspecific.

Historically, Miocene and Pliocene assemblages from the Northern Hemisphere have been investigated more intensively than those from the south (Boessenecker, 2013; Brandt, 1873; Gol’din & Startsev, 2016; Gottfried, Bohaska & Whitmore, 1994; Hampe & Ritsche, 2011; Kohno, Koike & Narita, 2007; Oishi & Hasegawa, 1995a; Steeman, 2010; Van Beneden, 1882; Van Beneden, 1886; Whitmore Jr & Kaltenbach, 2008), presumably because of the greater number of local researchers. Yet, despite these efforts, northern neobalaenines have remained elusive, including from the highly fossiliferous assemblages of Belgium (Steeman, 2010), northern Italy (Bisconti, 2009), Japan (Oishi & Hasegawa, 1995a; Oishi & Hasegawa, 1995b), and both the eastern and western coasts of the United States (Boessenecker, 2013; Gottfried, Bohaska & Whitmore, 1994; Whitmore Jr & Kaltenbach, 2008). By contrast, comparatively limited work in the Southern Hemisphere has yielded specimens from Australia (Fitzgerald, 2012; this study), Peru (Bisconti, 2012) and, possibly, Argentina (Buono et al., 2014), ranging in age from 10–5 Ma.

Figure 6 Global occurrence and age of neobalaenine fossils.

(A) Map illustrating neobalaenine fossil localities, as well as the geographical distribution of extant Caperea marginata. (B) Simplified phylogeny showing the fossil record of the three major lineages of extant baleen whales. For neobalaenines, dark green and red denote Southern and Northern Hemisphere occurrences, respectively. Grey illustrates the total cetotheriid fossil record. The age of NMV P233333 is marked with an asterisk. (A) modified from Tsai et al. (2017), (B) from Fitzgerald (2012) and Buono et al. (2014), under a Creative Commons Attribution License. Drawings of extant whales by Carl Buell. L., Late; Pleist., Pleistocene; Plio., Pliocene; Qu., Quaternary.

Given this marked discrepancy, we suggest that the absence of Caperea in the Northern Hemisphere during the Miocene and Pliocene is a genuine phenomenon. Fossil neobalaenines remain rare in the north since they mostly, or perhaps exclusively, occurred there during the Pleistocene (Tsai et al., 2017). Sea levels at that time were markedly lower than today (Miller et al., 2005), leading to coeval marine deposits becoming eroded and/or inundated whenever the ice retreated. In the south, the rarity of neobalaenine fossils may reflect genuine biological scarcity or habitat restriction, but likely also a simple lack of research effort. We predict that, as more Miocene and Pliocene assemblages are studied, neobalaenines will continue to emerge primarily from southern field sites and collections.

Supplemental Information

Supplemental Information 1 3D model of the external surface of NMV P233333

Click here for additional data file.

Supplemental Information 2 3D model of the inner ear of Caperea sp. (NMV P233333)

Click here for additional data file.

We thank David P. Hocking for assistance with scanning, Tim Ziegler for access to specimens, Carl Buell for providing illustrations of extant whales, and Cheng-Hsiu Tsai, Robert W. Boessenecker and an anonymous reviewer for their constructive comments.

Institutional abbreviations

NMV Museums Victoria, Melbourne

Additional Information and Declarations

Competing Interests

Author Contributions

Data Availability

The authors declare there are no competing interests.

Felix G. Marx conceived and designed the experiments, performed the experiments, analyzed the data, prepared figures and/or tables, authored or reviewed drafts of the paper, approved the final draft.

Travis Park performed the experiments, analyzed the data, prepared figures and/or tables, authored or reviewed drafts of the paper, approved the final draft.

Erich M.G. Fitzgerald conceived and designed the experiments, analyzed the data, authored or reviewed drafts of the paper, approved the final draft.

Alistair R. Evans performed the experiments, contributed reagents/materials/analysis tools, authored or reviewed drafts of the paper, approved the final draft.

The following information was supplied regarding data availability:

This article focuses on the description of a new fossil, accessioned as specimen number P233333, at Museums Victoria, Melbourne, Australia.

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
