# Peer review of "A Miocene pygmy right whale fossil from Australia"

_PeerJ, doi:10.7717/peerj.5025_

## Round 0.1 · original submission · Minor Revisions

Dear authors,

I have accepted the decision of 'minor revisions' from the three reviewers.

In the text can you ensure that the nominal authority for each binomen occurs after its first instance, and that said reference is included.

Thank you for choosing PeerJ, and I look forward to receiving your revised submission.

·

Basic reporting

See the attached file.

Experimental design

See the attached file.

Validity of the findings

See the attached file.

Additional comments

See the attached file.

Reviewer 2 ·

Basic reporting

See general comment below.

Experimental design

See general comment below.

Validity of the findings

See general comment below.

Additional comments

I am glad to see this manuscript and was very pleased to be able to review it. This work will be of high interest to other cetacean paleontologist and neontologist. The discovery of additional specimens referable to Caperea and its relatives continue to add to the poorly known evolutionary history of this enigmatic mysticete. The specimen described here is an important addition not only to further understand the history of this group, but also highlights the richness and potential of the marine mammal assemblage at Beaumaris. The description of the specimen is very thorough and concise. All the relevant literature is cited carefully and discussed where appropriate. The results and comparisons are well done and summarized as is the discussion and relevance of the discovery.

The manuscript is well written and the figures are of very high quality and informative. I specially appreciate the authors adding a side-by-side Figure comparing NMV P233333 with Caperea marginata. I do have some comments that are outlined in detail below and that I hope the authors take into consideration.

General comments:

1) Line 19: here and in several other instances the authors use “Late Miocene”, however being that late is not an internationally recognized division of the Miocene, I suggest they change to “late Miocene”.
2) Along those same lines, I suggest the authors also use “Messinian” as it is the recognized stage of the late Miocene relevant to the specimen they describe. e.g. line 88, change “… latest Miocene in age,…” to “… latest Miocene (Messinian) in age,…”
3) Although it doesn’t seem to be required by the journal, I suggest the authors add the taxonomic authority (and reference) at the first use of a taxonomic name.
4) Lines 35-36, mention the age of the fossils from Argentina and Australia.
5) Line 110: “aligned with the proximal portion of the facial canal” – please label facial canal in Figure 2C
6) Line 111: “The crista transversa” – please label in Figure 2C
7) Figures: If possible, I suggest the authors adding a side-by-side figure of NMV P233333 with Caperea marginata in dorsal view, even if they do as supplementary figure.

·

Basic reporting

The structure and language in this article is impeccable as usual; most data is contained within the main text of the manuscript. I have a few additional suggested references, and I suggest that the authors consider making the CT data and/or STL files available as SI.

Experimental design

The methods involved in the study are appropriate and adequately described in the text.

Validity of the findings

The authors have adequately substantiated their identification of the specimen.

Additional comments

The manuscript by Marx et al. is a short and sweet addition to the rapidly expanding body of literature on fossil baleen whales, reporting a partial periotic of a pygmy right whale from the uppermost Miocene locality of Beaumaris in Australia. Another specimen of a neobalaenine (or neobalaenid as most still prefer) was identified in an earlier study by Fitzgerald (2012) which was an isolated but highly distinctive cone-shaped composite posterior process of the petrosal and bulla. This specimen was identifiable only to the Neobalaeninae/Neobalaenidae. This new specimen preserves the body of the petrosal and the pars cochlearis, which is more diagnostic and thus the specimen is identifiable at the genus level - specifically owing to the very narrow constriction at the base of the anterior process (but see below). I have only a few moderate and a few additional minor comments, which I think the authors can easily address. Overall it is a fine piece of work and I heartily recommend acceptance after minor revision.

Kind regards, Robert W. Boessenecker, Ph.D.

Moderate: Given the high degree of abrasion and polishing of the specimen, how confident can the authors be in their interpretation of the three features identifying this specimen as Caperea (open facial canal, short posterior cochlear crest, small neck of anterior process)?

Provided the authors can address the above concern, an additional selling point is the fact that now two neobalaenine genera appear to have been present during the late Miocene, pulling the Caprea/Miocaperea split further back and precluding a simple ancestor/descendant relationship between Miocaperea and extant Caperea.

Additionally: I request that the authors comment on the identification of the earlier specimen reported by Fitzgerald (2012). Could it represent Caperea? Do you think it should be tentatively referred to Caperea sp. now that a more diagnostic specimen is known? I understand that if identifiable only to the subfamilial level it leaves open the possibility that two neobalaenines were preserved at Beaumaris, but according to present evidence parsimony would suggest that only one was present in the assemblage.

Minor comments:

Line 30: A citation to Boessenecker and Fordyce 2015 (Tokarahia: ZJLS) is highly relevant as it includes the largest morphological character matrix ever attempted for mysticetes and recovered a monophyletic Balaenoidea.

Line 36: suggest 'uncertainly identified past the level of Neobalaeninae"

Line 52: focus stacked

Line 109: What is the dorsal vestibular area? Please clarify.

Line 119: Please provide some examples. Is this the same feature preserved in some balaenids and certain species of "Kellogg's cetotheres" like Dioroctus/Aglaocetus? If so, there was a character state for one of Boessenecker & Fordyce's characters, though I cannot recall the exact number.

Line 122: shelf like process - this feature is not labeled. Please clarify.

---

## Round 0.2 · Minor Revisions

Dear authors,

I have made the decision of ‘minor revision’ as I would like to give you the chance to respond to the comments of reviewer one. Otherwise, I will be happy to choose ‘accept’.

I look forward to receiving your revised manuscript.

·

Basic reporting

Satisfactory.

Experimental design

Satisfactory.

Validity of the findings

Satisfactory.

Additional comments

This paper is more or less ready to go – just two more additional notes.

1. Measurements
There is indeed no standard set of periotic measurements. But, descriptions of new baleen whales with well-preserved periotics generally provided a wide range of measurements, for example, Boessenecker and Fordyce 2015 PeerJ: table 2, or Tsai and Fordyce 2015 Journal of Mammalian Evolution: table 2, which some measurements could be applied to NMV P233333 and then should give readers a better idea for visualizing the specimen.

2. CT data/STL files
Alternatively, if possible, it would be extremely helpful to provide the CT data or STL files in the Supplementary Information, which should allow readers and other researchers to calculate any measurements they desire and facilitate further comparative work, as Reviewer 3 also suggested.

Regards
Tsai

Reviewer 2 ·

Basic reporting

See general comments below.

Experimental design

See general comments below.

Validity of the findings

See general comments below.

Additional comments

The authors have addressed most of my earlier comments and have explained in detail when they disagreed. The finding reported in this manuscript are a valuable addition to our knowledge of the evolutionary history of cetaceans, specially neobalaenines. The figures and illustrations are excellent and very informative. I commend the authors on a well written manuscript and I am looking forward to seeing this work published.

---

## Round 0.3 · accepted · Accept

Dear authors,

Thank you for swiftly answering reviewer one's comments from the previous round. As such, I am happy to follow their decision of ‘accept’.

Once again thank you for choosing PeerJ as your publishing venue, and I hope you use us again soon.

#